# A Review of Bolt Tightening Force Measurement and Loosening Detection

**DOI:** 10.3390/s20113165

**Published:** 2020-06-02

**Authors:** Rusong Miao, Ruili Shen, Songhan Zhang, Songling Xue

**Affiliations:** 1Department of Bridge Engineering, Southwest Jiaotong University, Chengdu 610031, China; miaorusong@my.swjtu.edu.cn (R.M.); xuezhiying@my.swjtu.edu.cn (S.X.); 2Department of Civil Engineering, College of Architecture and Environment, Sichuan University, Chengdu 610065, China; songhan.zhang@scu.edu.cn

**Keywords:** bolt loosening, structural health monitoring, acoustoelastic effect, contact interface stiffness, nonlinear dynamics, impedance, pattern recognition

## Abstract

Pre-stressed bolted joints are widely used in civil structures and industries. The tightening force of a bolt is crucial to the reliability of the joint connection. Loosening or over-tightening of a bolt may lead to connectors slipping or bolt strength failure, which are both harmful to the main structure. In most practical cases it is extremely difficult, even impossible, to install the bolts to ensure there is a precise tension force during the construction phase. Furthermore, it is inevitable that the bolts will loosen due to long-term usage under high stress. The identification of bolt tension is therefore of great significance for monitoring the health of existing structures. This paper reviews state-of-the-art research on bolt tightening force measurement and loosening detection, including fundamental theories, algorithms, experimental set-ups, and practical applications. In general, methods based on the acoustoelastic principle are capable of calculating the value of bolt axial stress if both the time of incident wave and reflected wave can be clearly recognized. The relevant commercial instrument has been developed and its algorithm will be briefly introduced. Methods based on contact dynamic phenomena such as wave energy attenuation, high-order harmonics, sidebands, and impedance, are able to correlate interface stiffness and the clamping force of bolted joints with respective dynamic indicators. Therefore, they are able to detect or quantify bolt tightness. The related technologies will be reviewed in detail. Potential challenges and research trends will also be discussed.

## 1. Introduction

Bolts are widely used in buildings, bridges, machines and vehicles (Figure 1). Providing bolts with proper preload is crucial to guarantee the stability and safety of the bolted connection. Insufficient preload may lead to a slip between connected parts and overload may lead to strength failure of the bolt or local damage of structural components. Both of these cases are harmful to structures [1]. As referred to in the Chinese industrial code JGJ 82-91 (1992) [2], the variation of assembly fastening force for high strength bolts should be less than 10%.

In current practice, a single bolt is hardly tightened with an accurate load by using a torque wrench or hydraulic stretching jack due to the uncertain frictional influence on tools and nut threads. For a joint connection involving a number of bolts, it is difficult to tighten all the bolts at the same time, due to the limited number of devices or a lack of workspace. Instead, the bolts have to be tightened alternately for many passes in order to approximate the design load. This is due to many phenomena, but namely the interaction effect or the cross talk, which greatly increases difficulty in evaluating the final tension force locked in the bolts [3,4,5,6]. In the situation of bolted clamps installed on the main cable of suspension bridges, the nonlinear transverse compressed deformation of the cable under the action of the fastened clamp leads to a complex interaction effect on the bolts [7]. Authors have conducted a laboratory experiment to investigate the nonlinear interaction effect of the alternately tightening procedures on the cable clamp bolts, as depicted in Figure 2.

In addition to the initial difficulty in achieving the uniform bolt tension force in the assembling phase, the inevitable long-term loosening of the bolt over the in-service period also adds uncertainty regarding the quality of the joint. The mechanism by which bolts loosen is very complicated and primary factors include stress relaxation, plastic deformation and fretting of bolt threads. Service circumstances such as alternating load cycles, temperature, and lubrication coating also influence bolt looseness [8,9,10,11,12]. A previous survey investigated bolts loosening in the cable clamps of suspension bridges based on a large number of samples. This survey concluded that the bolts could self-loosen by up to 40% of their initial preload, thus causing a slip between the clamp and the cable which is quiet harmful to bridge safety [13]. Effective bolt monitoring techniques are required in order to retighten bolts for structural maintenance.

This paper reviews methodologies concerning bolt tightening force measurement and monitoring for loose bolts. The outline of this paper consists of the following parts. Section 2 introduces the torque wrench method, which is most commonly used in practice to control bolt preload. Section 3 presents state-of-the-art technologies of bolt load measurement based on the acoustoelastic principle. The algorithms adopted by the commercial instrument *MINI-MAX* bolt tension monitor, developed by Dakota Ultrasonics, is briefly introduced. Section 4 reviews active sensing approaches on the basis of contact dynamic phenomena, classified into two categories, linear and nonlinear. Section 5 provides an introduction about research on the impedance method. Other vibration-based methods are included in Section 6. Finally, conclusions and further comments are given in Section 7.

## 2. Torque Wrench Method

From previous studies, it has been confirmed that the relationship between the torque and the bolt preload is close to linear. Based on this, an empirical equation has been developed [14]:(1)$$T=\frac{F}{2}\left(\frac{d_{2}}{\cos\alpha }\mu_{s}+\frac{P}{\pi }+d_{w}\mu_{w}\right),$$
where T is the applied torque, F is the bolt preload, $d_{2}$ is the pitch diameter, α is half of the thread angle, $\mu_{s}$ is the frictional coefficient of screw threads, P is the pitch, $d_{w}$ is the equivalent friction diameter of the support surface of nut and $\mu_{w}$ is the frictional coefficient of the interface between nut and support surface.

The torque wrench method relies on the linear tension–torque relationship of bolt fastening. However, the tension–torque ratio is affected by a number of factors such as the coefficient of friction, thread tolerance errors and lubrication coat types. Therefore, it presents many discrepancies [15,16,17] and leads to tension force estimation errors of up to 25% [18]. This large error limits the application of the method especially when the accuracy of measurement is required. Research on the tension–torque relationship concentrates on the torque sharing problem. In order to reveal the mechanism of the torque sharing of bolts, theoretical models [19,20] and finite element models [21] were developed. Wang et al. [17] constructed tension–torque ratio testing apparatus for miniature bolts (M2) and portrayed time variation ratio value curves in the process of screwing the bolts with a wrench. In general, the torque wrench method is used in bolt fastening rather than in-service monitoring. The reliability and accuracy problem of this method is one of reasons developing in-service bolt tension measuring techniques is important.

## 3. Ultrasonic Measurement Based on Acoustoelastic Principle

The velocities of longitudinal and transverse plane waves, which propagate along the direction of the tested stress in an isotropic and homogeneous material, can be expressed only approximately by accounting for their first-order terms [22,23].
(2)$$\begin{array}{c} V_{\sigma }{}^{L}=V_{0}{}^{L}\left(1+A_{L}\sigma \right)\\ V_{\sigma }{}^{T}=V_{0}{}^{T}\left(1+A_{T}\sigma \right) \end{array}$$
where $V_{0}{}^{L}$ and $V_{0}{}^{T}$ are the longitudinal and transverse wave velocities of the unstressed state, respectively, $A_{L}$ and $A_{T}$ are the acoustoelastic constants of the longitudinal and transverse waves, respectively, σ is the bolt axial stress and $V_{\sigma }{}^{L}$ and $V_{\sigma }{}^{T}$ are the longitudinal and transverse wave velocities of the stress state of σ, respectively. Equation (2) represents the acoustoelastic principle, indicating that the increase of the tension force can be estimated through a predictable decrease of the wave velocities of the bolt. Specific measuring methods can be classified into two categories according to the type of utilized waves, either the single-type wave method or the multiple-type wave method.

### 3.1. Single-Type Wave Method

Algorithms applied by the *MINI-MAX* bolt tension monitor (which was developed by the company Dakota Ultrasonics) is based on the acoustoelastic principle [24]. By sending a longitudinal ultrasonic pulse at the end of a bolt, an echo signal reflected at the opposite end can later be detected. The ultrasonic length of the bolt can then be determined according to the time difference between the two signals, which is namely the time-of-flight (TOF). Elongation of the ultrasonic length of the bolt due to the tensile stress is defined as,
(3)$$X_{UE}=V_{0}\Delta t=V_{0}\left(t_{1}-t_{0}\right),$$
where $X_{UE}$ is the ultrasonic elongation, $V_{0}$ is the wave velocity of the material in an unstressed state, $t_{1}$ and $t_{0}$ are the TOF of a stressed state and the unstressed state, respectively, and Δt is the change of TOF. As the bolt tightens, the physical length of the bolt increases and the wave velocity decreases, both resulting in an increase of the ultrasonic length. Stress factor *S_F_* is introduced to measure the ratio of the physical elongation $X_{ME}$ to the ultrasonic elongation $X_{UE}$
(4)$$S_{F}=\frac{X_{ME}}{X_{UE}},$$
where $S_{F}$ is assumed to be constant because it mainly depends on the bolt’s material and typically has a relative error of less than 5% in total over the entire range of possible steel fasteners. Bolt load P is given by,
(5)$$P=X_{ME}\cdot L_{F}+L_{o},$$
where $L_{F}$ is the load factor constant and $L_{o}$ is the offset or error term to correct the measured tension. $L_{F}$ can be obtained according to Hooke’s law:(6)$$L_{F}=\frac{A_{\sigma }E}{L_{E}},$$
where $A_{\sigma }$ is the stress area of the bolt cross section, E is Young’s modulus and $L_{E}$ is the equivalent stressed length.

Figure 3 shows a calibration test (conducted by the authors) for the $S_{F}$ factor of the high strength bolts used in a cable clamp designed for a suspension bridge. The bolts were fabricated of 40 CrNiMoA steel, with a diameter of 52 mm and a length of 850 mm. The tests were implemented under a constant temperature. The linear relationship between the bolt load and the measured ultrasonic length is presented in Figure 3b. Similar linear relationships were validated by other research [25,26,27,28].

While ultrasonic length has been proven to be feasible to measure bolt tension, it has also been found that temperature has an evident effect on ultrasonic measuring results [29]. As temperature increases wave velocity decreases and the bolt thermally expands, which results in an increase of the TOF as well as ultrasonic elongation. The opposite is true when temperature decreases. Mathematical expressions in the frame of the acoustoelastic principle are given by:(7)$$V_{1}=V_{0}\left[1-k\left(T_{1}-T_{0}\right)\right]$$
(8)$$L_{1}=L_{0}\left[1+\alpha \left(T_{1}-T_{0}\right)\right],$$
where k is the temperature coefficient, α is the coefficient of thermal expansion and $T_{1}$ and $T_{0}$ are the temperature conditions. $V_{1}$ and $V_{0}$ are the wave velocities corresponding to the two temperature conditions, respectively, and $L_{1}$ and $L_{0}$ represent bolt length in the two temperature conditions, respectively.

The temperature-dependent thermal expansion/contraction and the stress-dependent wave velocity alternation of bolt belong to two independent physical natures which cause the same trend in changing the measured ultrasonic length. Therefore, in the *MINI-MAX* [24], the two effects are accounted for in a single factor, namely the temperature compensation coefficient ($T_{p}$) which is calibrated by:(9)$$T_{p}=\frac{L_{T}-L_{T_{0}}}{L_{T_{0}}\left(T-T_{0}\right)}\times 10^{6},$$
where T and $T_{0}$ are the two different measuring temperatures and $L_{T_{0}}$ and $L_{T}$ are the ultrasonic lengths at the two different temperatures. Therefore, when temperature largely differs for multiple measurements, temperature compensation is absolutely needed, which can be done using Equations (10)–(13).
(10)$$L_{U1}=V_{0}t_{0}-T_{p}\left(T_{1}-T_{0}\right)L_{T_{0}}\times 10^{-6};$$
(11)$$L_{U2}=V_{0}t_{1}-T_{p}\left(T_{2}-T_{0}\right)L_{T_{0}}\times 10^{-6};$$
(12)$$X_{UE}=L_{U2}-L_{U1};$$
(13)$$P=S_{F}\cdot X_{UE}\cdot L_{F}+L_{o},$$
where $T_{1}$ and $T_{2}$ are the two different measuring temperatures, $L_{T_{0}}$ is the unstressed ultrasonic length of the bolt at the reference temperature $T_{0}$, and $L_{U1}$ and $L_{U2}$ are the two measured ultrasonic lengths which are compensated at temperature $T_{0}$.

Although the algorithms based on the velocity change of the longitudinal ultrasonic wave are distinct in theory, techniques for precise measurement of the TOF, such as phase detection [26,30] and the high sampling rate for data acquisition systems (1.0 ns for 1 GS/s in an equivalent sampling rate), are required. The TOF in measured bolts is only in the range of tens of nanoseconds [31] which increases cost. The technique may not be reliable when the end of the bolt is rough, non-perpendicular to its axis (due to bending) or coated with thick paint. Couplant (coupling fluid) is needed to fill the small gap in the interface of the bolt end and the transducer to ensure adequate contact. This also produces uncertain measuring errors due to unknown and uncertain thickness of the couplant layer [32]. Apart from measuring the bolt purely under tensile action, the measurement based on the longitudinal ultrasonic wave for the bolt under comprehensive actions of shear load and tensile load has also drawn much attention [33,34].

### 3.2. Multiple-Type Wave Method

From the single-type wave method previously introduced, the change of bolt tightening force can be measured rather than the absolute value. In order to obtain the absolute value of the bolt tightening force directly, multiple-type wave methods—according to the velocity ratio between longitudinal and transverse ultrasonic waves—were developed [35,36].

#### 3.2.1. Use Longitudinal and Transverse Waves Separately

Pan et al. [37] derived a bolt stress function containing parameters including the velocities and TOFs of longitudinal and transverse ultrasound and Lame’s coefficients of materials. First, a tensile test is required to portray the curve of a pair of the intermediate variable composed of the TOFs and the bolt stress. Then, the rest of the coefficients can be solved using the least squares method, and the stress can be solved through this function. Experiments were conducted to ascertain the value of the coefficients in the different intervals of stress for low-carbon steel 4.8 and 8.8 bolts. Carlson and Lundin [38] verified that the ratio of TOFs between the transverse and longitudinal ultrasound varies linearly with the torque applied on a 1.1-m rock bolt. A standard cross-correlation technique using the signal envelopes was introduced to improve the precision of the estimated TOFs between the first and second echo.

#### 3.2.2. Use Mode-Converted Waves

Although the ratio of the velocities is feasible to measure the absolute value of the bolt tightening force, it should be noticed that technical issues exist for the transducers to simultaneously excite and receive both longitudinal and transverse ultrasonic waves. For the piezoelectric acoustic transducer (PZT), usually fabricated of lead zirconate titanate, the energy of a shear (transverse) wave may be too weak if the coupling fluid is not viscous enough. For the electromagnetic acoustic transducer (EMAT), on the other hand, the pure longitudinal wave of signal cannot be properly received because the Lorentz force does not function in generating a longitudinal wave in ferromagnetic materials [39,40,41].

In order to solve the above issues, new methods were proposed by utilizing the mode conversion of ultrasonic waves. When two orthogonally polarized shear waves strike the surface of a solid or liquid obliquely, they convert to longitudinally polarized waves. When two orthogonally polarized shear and longitudinal incident waves strike the surface obliquely, they convert into a longitudinal wave which is known as the mode conversion phenomenon [42].

Kim and Hong [42] added a spherical acoustic lens between the transducer and a bolt head to induce the mode-converted longitudinal incident ultrasonic wave (LL mode) refracted from the front interface of the bolt head and the mode-converted shear wave (LT mode) reflected from the rear interface of the bolt. Figure 4 shows the ray path of a mode-converted ultrasonic wave in a 15 mm length bolt when the acoustic lens is used. The wave LT is subjected to both longitudinal and transverse wave velocity, therefore producing a delay relative to the wave LL. Two echoes are easy to distinguish in the time domain. Theoretical expression of the bolt stress is given by,
(14)σ=ξ−ζγ,
(15)$$\gamma =\frac{TOF_{LL}}{TOF_{LT}},$$
where γ is the TOF ratio of the mode-converted waves LL and LT and ξ and ζ are the material constants. The linear relationship between the tensile stress σ and the TOF ratio γ was revealed in an experiment conducted on a 160 mm bolt. Nevertheless, the TOF ratio γ can be calculated only when the unknown refracted angles as shown in Figure 4 are determined by measuring TOFs in the couplant medium. Specific ray analysis renders the calculation complex and increases error source.

The EMAT generates and detects ultrasound through the magnetostriction without requiring the contact or the couplant for tested specimens [43]. This non-contact measure technique for bolt tension has been explored and has shown good performance [44]. Ding et al. [45] also described the bolt stress as a linear function of the TOF ratio of the mode-converted waves and the material constants. The TOFs of the mode-converted waves also need to be solved based on the ray path analysis, while the refracted incidence angle is an intrinsic constant depending on the EMAT. The linear relationship between the tensile stress and the TOF ratio was verified by an experimental test implemented on a 165 mm length M24 bolt.

Through the above reviews, the multiple-type wave methods have been proved theoretically and experimentally to be applicable for obtaining absolute bolt stress value. However, the effect of temperature on stress estimation through the multiple-type wave methods remains unclear, which is worth further study. In addition, the bending of bolt, especially of long bolt, yields derivations in the incidence angle which is used to solve the TOF ratio, thereby also affecting the stress measurement. This remains an open question so far, and further investigation is expected.

## 4. Contact Dynamic Method through Active Sensing

From a microscopic point of view, the solid surface is rough and uneven and real contact occurs only at some asperity surface peaks as illustrated in Figure 5. It is known that contact dynamic characteristics are influenced by the contact condition such as the real contact area and the contact stiffness, which are closely related to the clamping force of bolts [46,47,48,49,50,51]. PZT transducers can be used as actuators and sensors to detect and quantify the tightness of bolted joints according to the change of the contact dynamic responses. The approaches based on this concept can be divided into linear and nonlinear.

### 4.1. Contact Acoustic Linearity: Energy Attenuation Method

The wave energy attenuation (dissipation) method belongs to the linear method because the theory of the method assumes that the contact stiffness (i.e., the extent to which the interface resists deformation in response to a contact force) of the bolted connection system is linear [11]. Figure 5 illustrates the elastic wave propagation in a bolted joint. When the incident elastic waves carrying energy ($\Omega_{incident}$) propagate through the connected area of the bolted joint, some waves carrying energy ($\Omega_{leak}$) will leak through the real contact area (S) and the rest of the waves carrying energy ($\Omega_{transmitted}$) continuously transmit in the original connection member. The energy attenuation method assumes that the leaky or dissipated energy of the waves is proportional to the true contact area ($\Omega_{leak}\propto S$). According to the classical Hertz contact theory, the true contact area *S* and the contact pressure *P* have a relationship $S\propto P^{2/3}$. Thereby, the attenuated energy of elastic waves correlates with the contact pressure (i.e., the bolt clamping force) [11]. Yang and Chang [52,53] experimentally validated the energy attenuation phenomenon of the transmitted waves in a prototype apparatus and in a real bolted joint used to connect C–C composite thermal protection plates to the host structure. In order to measure the wave energy, Amerini and Meo [54] developed the first-order acoustic moment index $M_{0}$, defined as:(16)$$M_{0}=\int_{0}^{f_{N}}W\left(f\right)df,$$
where f is the frequency variable, $f_{N}$ is the Nyquist frequency and W(f) is the power spectral density (PSD) function of the received spectrum. Experimental studies indicate that high sensitivity of the sensor is extremely necessary for collecting high signal-to-noise ratio signals over a broad bandwidth, so every frequency component containing sensitive pressure-dependent energy can be caught. The technique of wavelet packet analysis was used to quantify the energy of the transmitted signal between the PZT patches [55,56]. Wang et al. [57] constructed an analytical model in which the attenuated energy is a function of the tangential damping and the normal bolt preload of the bolted joint, based on the fractal contact theory [58], and takes the imperfect contact interface into account. The center frequency of the emitted signal affects the received wave energy.

In recent years, a time reversal (TR) technique has been used in many fields due to its function of having a self-adaptive focus and obtaining a high signal-to-noise ratio of signals. The TR principle was introduced from optical applications to acoustics by Fink [59] and Ing and Fink [60]. The process of the TR can be described through Figure 5. Firstly, the PZT1 emits the input signal x(t)=Aδ(t), where δ(t) is a unit pulse and A is the amplitude of the pulse. Supposing the bolted joint is a linear time-invariant (LTI) system, the received signal by the PZT2 can be written as,
(17)y(t)=x(t)∗h(t)=A⋅h(t),
where “∗”denotes the convolution, “⋅”denotes the dot product and h(t) is the impulse response function (IRF) of the system between the source (i.e., actuator) and the sink (i.e., sensor). The response signal is then reversed in the time domain and results in,
(18)y(−t)=A⋅h(−t).

The reversed signal y(−t) is re-emitted as an input signal by the PZT2, and the focused signal is received by the PZT1:(19)$$\begin{array}{ll} y_{TR}\left(t\right) & =Ah\left(-t\right)*h\left(t\right)\\ & =A\int_{-\infty }^{\infty }h\left(\tau \right)h\left(\tau -t\right)d\tau \\ & =AR_{h}\left(t\right) \end{array}$$
where $R_{h}\left(t\right)$ is the autocorrelation function of h(t), which gets its maximum value at the time point t=0.
(20)$$y_{TR}{}_{\max}\left(t\right)=y_{TR}\left(0\right)=A\int_{-\infty }^{\infty }h^{2}\left(\tau \right)d\tau .$$

Substituting Equation (17) into Equation (20) yields the following:(21)$$y_{TR}{}_{\max}\left(t\right)=\frac{1}{A}\int_{-\infty }^{\infty }y^{2}\left(t\right)dt.$$

Equation (21) indicates that the energy of the response signal is proportional to the peak amplitude of the focused signal. Therefore, the energy of the response signal can be characterized by the peak amplitude of the focused signal. The experimental results demonstrate that the TR method is superior to the direct energy method in aspects such as consistency, sensitivity, and anti-noise properties [61,62,63].

As the contact pressure of bolted connectors increases, the true contact area will constantly increase until reaching a maximum value known as the contact saturated state [47,57,64,65,66,67,68,69]. In the saturated state, the wave energy attenuation does not change notably. This means bolt loosening at the embryo stage, where the contact pressure of the bolted connectors is relatively high, is imperceptible by the wave energy attenuation. Therefore, the contact saturation phenomenon restricts the sensitivity of the energy attenuation method in detecting bolt loosening. Experimental [64] and numerical [69] studies were performed to investigate the relationship between the surface roughness and the saturated value of the focused signal energy. These studies concluded that greater surface roughness always leads to larger focused signal energy in the contact saturation state. The TR technique was also reported to monitor the tightness of a threaded pipe connection. However, large discreteness of the focused signal energy was observed when the pipe connection was intensively loosened [63].

### 4.2. Contact Acoustic Nonlinearity: High-Order Harmonics and Sidebands Methods

#### 4.2.1. Concept

It is well known that the frequency responses of linear systems are almost the same as the random excitations of the input. By contrast, for a nonlinear system, the frequency responses of new contents with respect to the input excitations are initiated [70]. Potential defects such as micro-cracks, delamination, and de-bonding that exist in structures have been substantiated to lead them with some level of nonlinearity. These defects can therefore be sensed through vibration data analysis to correlate the defects with those newly initiated frequency response contents. This issue has been of great interest in the field of structural health monitoring (SHM) for decades [71,72,73].

As a special fault case, weakly fastened bolted joint structures due to bolt loosening also leads the system to be nonlinear, whereas, a well-tightened (healthy) bolted structure mainly behaves like a linear system [71]. Taking advantage of this concept, vibration features based on contact acoustic nonlinearity (CAN) have been widely investigated to detect and quantify loose bolts. Among them, high-order harmonics and sidebands are the most popular vibration features being focused on. For the former, only a single probing input signal is needed as the excitor, and the evident response of which the frequency exceeding the maximum frequency of the input will be initiated when the bolt is loosening [74,75,76]. Thus, this feature is known as high-order harmonics. For the latter, the sidebands are a production of the vibration modulation. To perform a modulation, bitonal excitations with two differentiated frequency bands are required to excite the bolted joint simultaneously. The low frequency (LF) one is an impact or pumping vibration which generates a modulated signal. The high frequency (HF) one functions as the probing wave (i.e., the carrier signal). The LF excitation compels the imperfect contact surface (due to the looseness of the bolt joint) to open and close periodically, similar to a breathing motion. This perturbs the HF probing wave and generates two peaks of frequency response in the spectrum located at the left and right side close to the probing wave, namely the left sideband and the right sideband, respectively [77,78]. A similar mechanism is also used to detect the “breathing” fatigue cracks [79,80,81].

To gain a physical understanding of the above concept and to correlate the nonlinear vibration features with the clamping force of the bolt (also referred to as contact pressure), many mathematical models including single degree of freedom (SDOF) systems and multi-degree of freedom systems have been established [82,83,84]. In the frame of these models, the interface stiffness of the bolted joint is treated as some kind of nonlinear spring, which is an essential source of the nonlinear vibration feature. Generally, the nonlinear stiffness of the spring is assumed to be quadratic nonlinearity corresponding to the freedom, based on physical reasoning [83]. Moreover, it is depicted to increase with a decrease in contact pressure and it also follows a power law [46]. Under this assumption, the amplitude of the sidebands and high-order harmonic will be designed to increase as the contact pressure decreases.

#### 4.2.2. Methodology

In order to effectively measure high-order harmonics and modulation sidebands and to utilize them to infer joint tightness, many damage indices have been developed and examined through both theoretical models and experimental tests. The fundamental manner is to apply the dimensionless indices in which the magnitude of the higher frequency component is in ratio to that of the lower fundamental frequency component, thereby eliminating the influence of the excitation amplitude. For example, Amerini et al. [82] applied a bandwidth filter at the first and second natural harmonic frequency to isolate linear and nonlinear parts of the spectrum. They then used a damage index given by the ratio between the PSD of the second harmonic filtered signal and the PSD of the maximum amplitude of the fundamental frequency filtered signal to evaluate the bolt loosening. Experimental result showed that the PSD ratio index outperforms the direct signal energy index with less residual error in the fitted curve.

Apart from the displacement of the joint, vibration data of strains and their time derivatives were also used to indicate joint tightness [85]. Milanese et al. [85] deployed four fiber Bragg grating strain sensors on a composite beam, bolted at both ends to steel support plates. A broad-band Gaussian signal was input into the beam. The root mean square (RMS) ${\bar{P}}_{\left[0,f_{\mathrm{thr}}\right]}$, between 0 and maximum excitation frequency, and ${\bar{P}}_{\left[f_{\mathrm{thr}},f_{\mathrm{Nyq}}\right]}$, up to the Nyquist frequency, indicate the average power contained in the two frequency bands. The damage index was defined as:(22)$$R=\frac{{\bar{P}}_{\left[f_{\mathrm{thr}},f_{\mathrm{Nyq}}\right]}}{{\bar{P}}_{\left[0,f_{\mathrm{thr}}\right]}}.$$

Both theoretical and experimental results revealed that compared with the raw data of the strain, their first and second time derivatives enhanced the high frequency differences which represent the nonlinearity of the system, thus being more sensitive to the change of joint tightness.

In the theory frame of the relationship between the fundamental and second harmonic amplitudes denoted by $A_{1}$ and $A_{2}$, respectively, in a solid–solid contact interface, the ratio $A_{2}/A_{1}{}^{2}$ was derived to be independent of the input amplitude [46]. Experimental tests performed in [86] examined that the ratio $A_{2}/A_{1}{}^{2}$ was approximately constant (of variation ±6.6%) under different excitation magnitudes. In addition to the nonlinearity induced by weak contact pressure, Yan et al. [86] also quantitatively studied the influence of other experimental sources of nonlinearity, including material, transmitter transfer function, and coupling on the measured results. Based on the experiment performed on a metallic and a composite bolted joint, Zhang et al. [84] systematically investigated the vibro-modulation method (i.e., the sidebands) and the second and third harmonic method to indicate joint tightness and sensitivities to the excitation magnitude. The main damage indices proposed in [84] are as follows:(23)$$\beta_{{}_{SOH}}^{Theory}=\frac{A_{SOH}}{A_{LF}^{2}};$$
(24)$$\beta_{{}_{TOH}}^{Theory}=\frac{A_{TOH}}{A_{LF}^{3}};$$
(25)$$\begin{array}{c} \beta_{{}_{VAM}}^{Theory}=\frac{1}{2}\left(\beta_{LS}^{Theory}+\beta_{RS}^{Theory}\right)\\ =\frac{1}{2}\left(\frac{A_{LS}+A_{RS}}{A_{LF}A_{HF}}\right) \end{array}$$
where $A_{SOH}$ and $A_{TOH}$ are the amplitudes of second order and third order harmonic, respectively. $A_{LF}$ and $A_{HF}$ are the amplitudes of the LF pumping force and the HF probing force, respectively. a $A_{LS}$ and $A_{RS}$ are the amplitudes of the left sideband and the right sideband, respectively. $\beta_{{}_{SOH}}^{Theory}$ and $\beta_{{}_{TOH}}^{Theory}$ represent the magnitudes of second and third order harmonic independence of $A_{LF}$ and $\beta_{{}_{VAM}}^{Theory}$ represents the average magnitude of the left and the right sideband independence of $A_{LF}$ and $A_{HF}$. The experimental results showed that $\beta_{{}_{SOH}}^{Theory}$ and $\beta_{{}_{VAM}}^{Theory}$ were insensitive to $A_{LF}$ and $A_{HF}$ and that $\beta_{{}_{TOH}}^{Theory}$ was insensitive to $A_{HF}$, whereas $\beta_{{}_{TOH}}^{Theory}$ was sensitive to $A_{LF}$. Although the effort in the above literature is greatly helpful to utilize the CAN-based indices in inferring joint tightness, other testing parameters including probing force frequency and location of excitations and sensors can also distort testing results [83].

Recently, a novel nonlinear second-order output spectrum (SOOS) approach was proposed and was theoretically and experimentally validated to detect and locate bolt loosening faults in a ring-type structure with higher sensitivity and effectiveness than the traditional second-order harmonic-based method [87,88]. The SOOS is computed by a novel nonlinear spectrum decomposition method [89] which is based on the Volterra series, thus receiving the least coupling effects from other inherent nonlinearity (more than two order). The decomposition method can also be used to select optimized excitation magnitude in order to achieve the least estimation error for the SOOS [87].

A novel equipment-free bolt loosening detection approach through audible modulation was proposed in [90]. Through testing, the bitonal excitations from two PZT actuators with the carrier frequency of 117.8 kHz and the modulated frequency of 12,320 Hz were applied to a bolted plate and a bolted washer. Once the bolt was loose, an audible beep sound with the frequency of 12,320 Hz would be easy for the operator to hear. Therefore, there is no need for extra signal processors.

To conclude, a great number of theoretical and experimental studies in the literature prove that the CAN phenomena are sensitive to the tightness of bolted joints. High-order harmonic-based and VM sideband-based damage indices generally increase as the contact pressure decreases and follow the power law [55,82,83,84,86]. Since useful frequency range of excitations is large and relatively lower (compared to ultrasonic methods), sensors and transducer devices could be more flexibly chosen to accomplish the monitor task which would be more economically efficient. However, most studies in the current literature were based on lab experiment set-ups and relatively simple models and may not be effectively applied to real complex structures. The main challenges are (a) many test parameters are sensitive to the measured data; (b) there is an unfavorable and uncertain influence of damping and noise on the higher frequency component; (c) initial bolt loosening is insensitive to the CAN-based indices; (d) there are different natures of nonlinearities (bolt loosening, inherent defect like crack) coupled in the vibration features.

## 5. Impedance Method

Another active sensing technique to monitor bolt tightness is based on impedance signatures. The PZT actuator–sensor patch is boned at the surface of tested bolted connectors to input an alternating voltage sweep signal, typically from several to hundreds of kilohertz, and electrical impedance is then recorded for analysis. The electrical impedance signature depends on the mechanical impedance of the system containing both host structure and the boned actuator–sensor patch [91,92,93]. In general, the loose bolt causes a reduction of contact stiffness and increases damping, which will lead the resonance peaks to shift to lower frequencies and the peak height to descend [94].

Ritdumrongkul and Fujino [95] used the spectral element method (SEM) to simulate the impedance response of an aluminum beam connected with multiple bolts. First, by minimizing the difference between the measured impedance spectra and the SEM simulation, the nominal spring stiffness and the damping parameters were determined. Then, the spectral element model was used to simulate a number of loose bolt cases including various loosening levels and locations. Finally, the recognition work of these cases was tried by matching the measured impedance spectra with the simulated ones. It was suggested that the PZT patch be preferentially located where it could produce many impedance peaks. For the bolt group installed on a frame beam, the results of the experiment conducted by Wang et al. [96] indicated that only the PZT patch was located close to the loosening bolts. The RMS deviation (RMSD) indices of the real part of admittance (i.e., the inverse of impedance) can evaluate the degree of loosening.

In recent years, development of PZT impedance-based wearable devices has drawn much attention. By embedding a PZT patch in a washer, a smart washer was developed to evaluate bolt loosening through impedance frequency analysis [97,98,99]. For instance, Wang et al. [99] designed and fabricated a wearable PZT impedance device to monitor the tightness of a 12-bolt subsea flange. As shown in Figure 6, a small aluminum plate boned with a PZT patch composed of a monitor device was developed by Huynh et al. [100]. The aluminum plate is comprised of two boned sections on two sides and one suspending section in the middle, which makes it able to produce a unique vibration. The device was used to evaluate the bolt tightness of a connected plate as shown in Figure 6. The electrical-mechanical impedance (EMI) function was derived. Using the software COMSOL Multiphysics, finite element simulation was performed to pre-ascertain the sensitive frequency band of the peak impedance of the tested system. The experimental results showed that the trend of resonance frequency and the real impedance peak value with respect to the loose bolt agreed with the analytical results.

## 6. Other Vibration Based Methods

Analytical analysis indicates that the natural frequency (mode) of the washer decreases as the clamping force decreases [101]. Under this concept, Hosoya et al. [102] suggested using local mode frequency of the bolt head to evaluate the clamping force. Experimental results verified the monotonic relationship between the bolt clamping force and the local mode frequency of the bolt head. The smaller bolt size (or bolt mass) requires a higher frequency region of excitation in order to catch the change of the local mode frequency when the bolt clamping force changes. For joints with multiple bolts, the position of the bolt increases the difficulty in using this method and more systematic studies are needed [102]. Based on a model of a prestressed one-dimensional beam with elastic supports at both ends, the boundaries of stiffness in terms of the bolt tension is related to the natural transverse frequency—as the bolt is gradually tightened, the squared normalized natural transverse frequency experiences nonlinear-to-linear change. Proven by experimental tests conducted in [103], this method is considered feasible in monitoring the loosening of in-service bolts.

Based on vibrational input and output data, pattern recognition techniques were employed to improve the efficiency of bolt loosening fault indication. For example, Kong et al. [104] used impulse hammer tapping of the bolt head in a horizontal direction. The frequency response spectrum was divided into three segments and the PSD of each segment was arranged as a scalar metric signature parameter. With the help of machine learning using the decision tree method, the PSD signature parameters derived from different tightness levels of the bolts was discriminated in the 3D Cartesian coordinate space, in which each segment represented a dimension. Li et al. [105] utilized the probability densities function (PDF) of two output signals of the continuous stationary vibration produced by a machine to implement principal component analysis (PCA) [106] and matrix fusion. The projection points corresponded to different bolt tightness degrees which could then be discriminated in 3D or 2D spaces. Combining the use of six time-domain parameters and 14 frequency-domain parameters extracted from the output vibration data produced by known impulse excitation on a bolted tunnel fan foundation, Chen et al. [107] established a series of preliminary indicators from which the sensitive indicators were first picked out by the scatter matrix [108]. Then, a number of traditional manifold learning algorithms were implemented to compress dimensions of the indicators aiming at improving their clustering performance and identification ability. Through comparison, the orthogonal neighborhood preserving embedding (ONPE) algorithm [109] was believed to function the best. Non-contact laser ablation is able to excite high frequency response, and it was used in [110] to excite the impulse response on a six-bolt joint cantilever. The Mahalanobis–Taguchi (MT) recognition method [111] was used to establish a statistical evaluation damage index based on the measured frequency response function (FRF) data. The experimental results and numerical simulations validated the feasibility of the method to detect and locate the loose bolt in the joint [110].

## 7. Conclusions

Bolted joints are important connections commonly used in many engineering situations. Proper bolt tightening force is a key factor in connection safety. However, it is difficult to tighten a group of bolts in a uniform and accurate design due to the interaction effect, and already tightened bolts will inevitably loosen in states of high stress. Therefore, effectively measuring bolt tension force and detecting and quantifying the tightness of bolted joints is of great importance. This issue belongs to the field of structural health monitoring, where bolt loosening is regarded as a type of damage or fault that widely exists in bolt-connected structures. The technical summary of the methods discussed in this paper is briefly listed in Table 1.

Based on the review of the literature, the following conclusions and discussions can be drawn.

The ultrasonic measurement based on the acoustoelastic principle is able to quantitatively measure stress levels for tightened bolts. This basic technique is done by measuring the TOF of a single longitudinal ultrasonic wave, and a commercial instrument has already been invented. However, the single-type wave method is unable to obtain an absolute value of the stress retained in an in-service bolt, unless the unstressed ultrasonic length of the bolt is known. The multiple-type wave method takes advantage of the linear relationship between bolt axil stress and TOF ratio of the longitudinal wave to the transverse wave. This method can obtain the absolute value of bolt stress, however it is more complicated and has a high error of sources. The influence of temperature, couplant, bolt bending and wave path on the measurement accuracy of the ultrasonic measurement methods, especially for the multiple-type wave method, is valuable.The contact dynamic method through active sensing evolved from the techniques applied to the inherent defect (fault) detection of solid, which is of great interest in the SHM field. Supported by rich proof of the experimental, theoretical, and numerical research results, indicators based on wave energy attenuation and nonlinearities are feasible to detect bolt loosening and quantify the tightness of bolted joints. However, they are easily affected by test parameters such as probing force frequency and the location of excitations and sensors and must therefore be controlled for. Compared to ultrasonic methods, the effective exciting and sensing signal frequency of the contact dynamic methods has a lager and lower range, which means it could be a more economically feasible option. Since the most current studies are based on relatively simple and concept-proof theories and experimental set-ups, more studies are needed to improve the sensitivity and robustness of the methods when they are applied to real and complex cases.The impedance method is based on the relationship between the bolt tightening force and the shift of resonance frequency and its peak value of the joint. Other vibration-based methods correlate the local mode frequency of bolted joints or the signature parameters from PDF or FRF analysis to bolt clamping tightness. Some pattern recognition techniques were employed to improve the identification ability of these indicators. In order to improve the sensitivity of the vibration-based methods, the technical issues of accurate recognition and measurement of high-frequency local dynamic responses still deserve further research. In addition, studies on the design manner of smart wearable tightness monitoring equipment suiting diverse bolted joint types are expected.

## Figures and Tables

**Figure 1 sensors-20-03165-f001:**
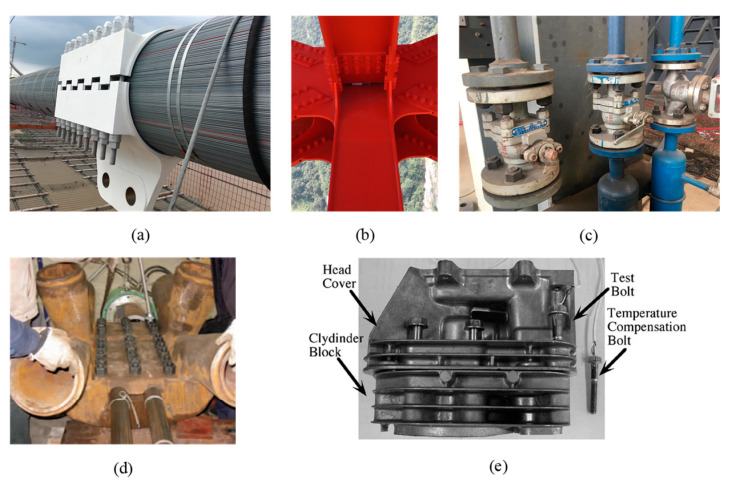
Applications of prestressed bolted connections. (**a**) Main cable clamp in a suspension bridge. (**b**) Steel truss structure. (**c**) Flange. (**d**) Joint of cable-supported structure [6]. (**e**) Engine head joint [8].

**Figure 2 sensors-20-03165-f002:**
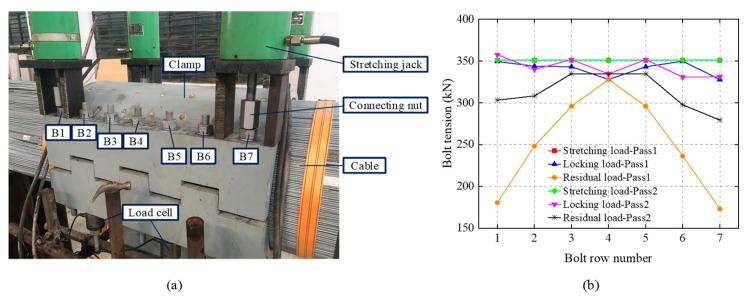
Interaction effect of the cable clamp bolts of a suspension bridge in an alternately fastening process. (**a**)Experimental set-up. (**b**) Alternation of tension force of the bolts during two passes of fastening from sides (bolt B1 and B7) to middle (bolt B4).

**Figure 3 sensors-20-03165-f003:**
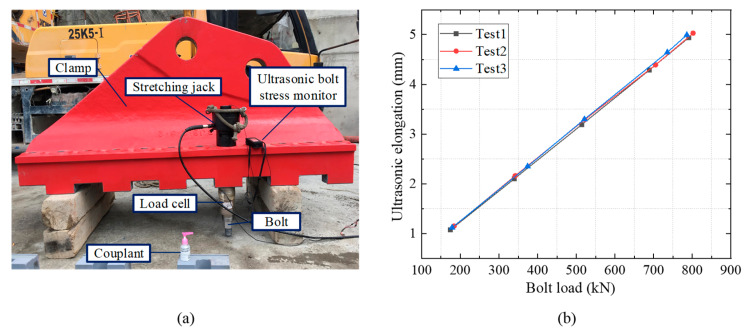
Field calibration test for ultrasonic stress factor. (**a**) Test set up and (**b**) relationship between bolt load and measured ultrasonic elongation.

**Figure 4 sensors-20-03165-f004:**
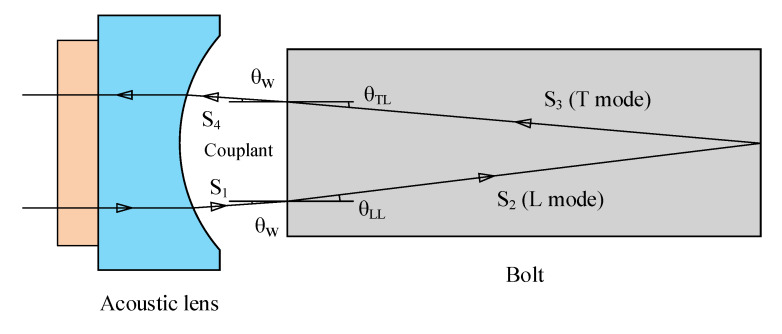
Schematic of the ray path of a mode-converted ultrasound (LT mode) in a short bolt (15 mm long).

**Figure 5 sensors-20-03165-f005:**
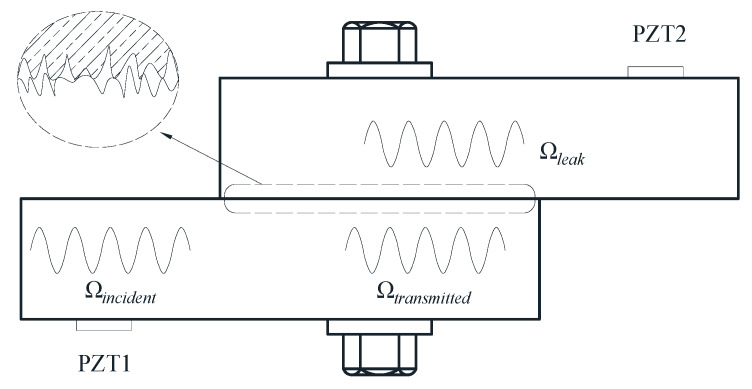
Schematic diagram of the elastic wave energy attenuation in the interface of a bolted joint.

**Figure 6 sensors-20-03165-f006:**
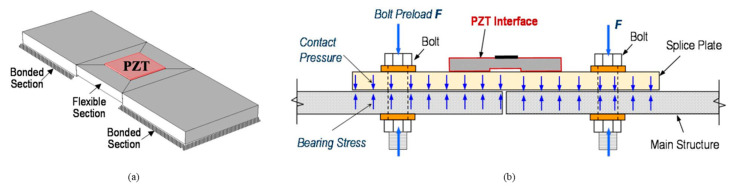
Impedance monitoring method for bolted connection via a piezoelectric acoustic transducer (PZT) interface. (**a**) PZT interface and (**b**) bolted connection equipped with a PZT interface [100].

**Table 1 sensors-20-03165-t001:** Technical summary of methods.

Methods	Principles or Measurement Tools	Merits	Challenges	Application Cases
Ultrasonic measurement method	Acoustoelastic principle	Able to quantify the axial stress of bolt	Obtain the absolute value of bolt stress	Commercial application case
Contact dynamic method	Wave energy attenuation or contact acoustic nonlinearity (CAN)	Broader and lower useful excitation frequency range; relatively simple and flexible devices	Robustness and sensitivity	Simple theory models and proof-of-concept experimental set-ups
Impetance method	Impedance spectrum	Flexible devices; wearable equipment design	Robustness and sensitivity	Simple theory models and proof-of-concept experimental set-ups
Other vibration based methods	frequency response function (FRF), power spectral density (PSD), probability densities function (PDF), etc.	relatively simple and flexible devices	Robustness and sensitivity	Simple theory models and proof of principle experimental set-ups

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
