# Peer review of "A Review of Bolt Tightening Force Measurement and Loosening Detection"

_sensors, 2020, doi:10.3390/s20113165_

Round 1
Reviewer 1 Report
The paper reviews state-of-the-art research on bolt tightening force measurement and loosening detection. It introduces the torque wrench method, bolt load measurement based on the acoustoelastic principle, MINI-MAX bolt tension monitor details, active sensing approaches on the basis of contact dynamic phenomena, research on the impedance method, and other vibration based methods. Finally, some conclusions and further comments are given together with 113 references. The main contribution of this review paper is to comprehend all recent methodologies for bolt tightening force measurement and loose monitoring at one place, and could serve as good basis for someone who would like to be more familiar with this field.
Author Response
We appreciate gratefully your positive comments on this paper.
Reviewer 2 Report
The paper presents a review of bolt tightening force measurement and in-services monitoring. The article is well written and covers the leading techniques and their theoretical background. The authors are invited to consider the following recommends:
- L121: As mentioned in the paper, the stress factor, SF is assumed to be a constant and considered as a material property. It will be beneficial to the readers if a discussion on the parameters affecting the stress factor is provided. Also, the sensitivity of the measurements to this factor needs to be discussed.
- L130: Add more detail about the bolts you have used for calibration.
- L182: What was the reason for not including the stress function (equation) in the manuscript?
- A technical summary of the techniques reviewed in the paper can be presented as a table to demonstrate advantages and disadvantages of each technique briefly.
- In the conclusion part, a discussion on the research gaps and recommendations for future research trends will be valuable.
Reviewer 3 Report
Comments to the Author
Review of the paper „A review of bolt tightening force measurement and loosening detection“
Written by Rusong Miao, Ruili Shen, Songhan Zhang, Songling Xue
This paper reviews the state-of-the-art research on bolt tightening force measurement and loosening detection, involving the fundamental theories, algorithms, experimental set-ups, and practical applications. It is interesting paper, it is generally well written and fits in the field of an international journal of Sensors.
Remarks:
The reviewed methods should be more compared. The article should reflect the advantages of the methods, disadvantages, measurement features, measurable parameters, accuracy and reliability of the methods. The compared results should be presented in table or discussion.
Line 101: Not all characters are described under the Equation (2).
In section 3.1 editorial corrections are needed.
Figure 4 should be of better quality.
Line 292: is it correct equation (21)?
Line 294: is it correct equation (22)?
Line 402 and 403: 117.8 kHz, 12,320 Hz point or comma? And rounding of numbers?
Line 465: to much spaces.
Conclusions:
This is an interesting paper. Such a paper could certainly interest readers of MDPI Sensors journal. Therefore, this paper is recomended for publication in MDPI Sensors journal with the suggested corrections in the review.
